

# An ensemble based approach using a combination of clustering and classification algorithms to enhance customer churn prediction in telecom industry

Syed Fakhar Bilal[1], Abdulwahab Ali Almazroi[2], Saba Bashir[1], Farhan Hassan Khan[3] and Abdulaleem Ali Almazroi[4]

[1] Computer Science Department, Federal Urdu University of Arts, Science and Technology, Islamabad, Pakistan
[2] University of Jeddah, College of Computing and Information Technology at Khulais, Department of Information Technology, Jeddah, Saudi Arabia
[3] Knowledge & Data Science Research Center (KDRC), Computer Engineering Department, National University of Science and Technology, Islamabad, Pakistan
[4] Department of Information Technology, Faculty of Computing and Information Technology, King Abdulaziz University, Rabigh, Saudi Arabia

Corresponding author
Saba Bashir,
saba.bashir@fuuast.edu.pk

## ABSTRACT

Mobile communication has become a dominant medium of communication over the past two decades. New technologies and competitors are emerging rapidly and churn prediction has become a great concern for telecom companies. A customer churn prediction model can provide the accurate identification of potential churners so that a retention solution may be provided to them. The proposed churn prediction model is a hybrid model that is based on a combination of clustering and classification algorithms using an ensemble. First, different clustering algorithms (*i.e.* K-means, K-medoids, X-means and random clustering) were evaluated individually on two churn prediction datasets. Then hybrid models were introduced by combining the clusters with seven different classification algorithms individually and then evaluations were performed using ensembles. The proposed research was evaluated on two different benchmark telecom data sets obtained from GitHub and Bigml platforms. The analysis of results indicated that the proposed model attained the highest prediction accuracy of 94.7% on the GitHub dataset and 92.43% on the Bigml dataset. State of the art comparison was also performed using the proposed model. The proposed model performed significantly better than state of the art churn prediction models.

## INTRODUCTION

Data mining is the way of identifying patterns and extracting knowledge from large amount of data (*Han, Pei & Kamber, 2011*). It allows us to identify the future flow employing different prediction models. Enterprises can predict future patterns and trends using different data mining tools and techniques. The purpose of data mining is to analyse

meaningful information that is hidden in huge amounts of datasets and incorporate such information for useful tasks (*Rustam et al., 2019*).

The telecom sector is becoming one of the most important industries in developed countries since the past two decades (*Ullah et al., 2019*). Data mining plays a vital role for prediction and analysis in the telecom industry due to availability of huge data. The basic application area is to perform prediction of churner in order to save customer retention and to make a high-profit rate. Data mining techniques are used in the telecom sector to observe the churn behaviour of the customers.

With the increasing rate in users of telecom companies, they now offer variety of services for the retention of customers. In order to obtain better services and benefits, the customer switches its service provider and the phenomenon is known as churn. If a customer switches a service provider's company then face loss occurs in the company's revenue. Prediction can be performed to identify the potential churners and retention solutions may be provided to them. A large number of mining algorithms are available which classify the behavior of customers into churner and non-churners.

The telecom sector is growing rapidly due to different technologies. Different companies provide quality of data for communication; some gives better services as compared to others. In order to stop churn, companies offer different services which are attractive for their customers. Data mining technologies are used to perform churn prediction using different algorithms like Naïve Bayes, decision tree, neural network and logistics regression *etc*. An accurate prediction model is very helpful for correct identification of customer's churn and plays a vital role in making decisions about their retention (*Vijaya & Sivasankar, 2018*). The best customer churn prediction model can identify churner and gives directions to decision-makers for generating maximum profit (*Höppner et al., 2017*; *Ali et al., 2018*; *Amin et al., 2019*).

There are number of reasons for which churning of customers. Most important of them are calls or packages rate which do not suit the customer (*Tiwari, Sam & Shaikh, 2017*; *Petkovski et al., 2016*). We can identify weather a customer wants to leave or not on the basis of his historical data and behavior.

Existing studies show that an efficient churn prediction model should efficiently use large volume of historical data in order to perform churner's identification. However, there are number of limitations in existing models due to which it is not possible to perform churn prediction efficiently and with high accuracy. A large volume of data is generated in telecom sector which contains missing values. Prediction on such type of data results in poor/inaccurate outputs for prediction models in literature. Data preprocessing is now performed to resolve this issue and missing values imputation is performed using machine learning methods which results in high performance and classification/prediction accuracy. Feature selection is also performed in literature; however, some important and information rich features are neglected during model development. Moreover, statistical methods are used for model generation which results in poor prediction performance. Furthermore, benchmark datasets are not used for model evaluation in literature resulting in poor representation of true picture of data. Fair comparison between different models

cannot also be performed without benchmark datasets. An intelligent model can be used to resolve the existing issues and to provide churn prediction more accurately.

The proposed churn prediction model is based on combination of clustering and classification algorithms. The performance of proposed model is evaluated on different churn prediction datasets. The evaluation of proposed churn prediction model is evaluated using different metrics such as accuracy, precision, recall and f-measure. The objectives of proposed research are; to identify the issues in literature and provide an efficient model for customer churn prediction, to identify the churners with high accuracy. The retention strategies may then be provided to the potential churners. It is also observed from the experiments that proposed churn prediction model performed better in terms of churn prediction by achieving high accuracy.

The main contributions of this research are as follows:

- Proposed a churn prediction model with high accuracy;
- Data preprocessing is performed for missing values imputation, noise removal and duplicates removal;
- Selection of important features using feature selection technique;
- Combination of clustering and classification techniques to perform customer churn prediction on two large datasets of telecom sector;
- Customer profiling is performed using clustering technique to divide the behavior of customer into different groups like low, medium and risky.

The remaining organization of paper is as follows: "Literature Review" provides literature review. "Proposed Methodology" presents the proposed churn prediction model. Experimental evaluation and results are presented in "Experimental Results, Evaluation and Discussion". Finally, "Conclusion and Future Work" presents the conclusion and future work.

## LITERATURE REVIEW

*Ullah et al. (2019)* proposed a churn prediction model named JIT-CCP model. In this model first step is data pre-processing, second step is binary classification and evaluation of performance by using the values of confusion matrix corresponding to true positive, true negative, false positive and false negatives. Based on these terms, probability of detection is calculated. Probability of detection (PD) is used to calculate the accuracy of multiple classifiers. If PD value is near to 1 then the classifier's results are much better and *vice versa*. However, the proposed model is not suitable for a large amount of data.

*Vijaya & Sivasankar (2018)* presented that customer retention plays a valuable role in the success of a firm. It not only increases company's profit but also maintains company's ranking among telecom industry. Customer retention is less costly rather than making new customers. So maintenance of the customers and customer association management (CAM) are the two parameters for the success of every company. In this research hybrid model of supervised and unsupervised techniques are used for churn prediction. There are different stages of this modal. In the first stage, data is cleaned and

removed different deviations from data. In next stage, testing and training sets of data are obtained from different clusters. After this, prediction algorithms are applied. In the final stage accuracy, specificity and sensitivity are measured for evaluating the efficiency of proposed model.

*Höppner et al. (2017)* stated that customer retention policies rely on different predictive models. The most recent development is expected to maximize the profit (EMPC) which selects the most valuable churn model. In this research a new classification method has been introduced, which integrates the (EMPC) matrix directly to churn model. This technique is known as ProfTree. The main advantage of this model is that a telecom company can gain maximum profit. The proposed model has increased performance and accuracy as compared to other models. In future this model may be combined into different algorithms to further increase the prediction performance.

*Ali et al. (2018)* used different mining algorithms and techniques for prediction of churners. WEKA software is used for applying different classifiers. The first step of this model is data preprocessing where missing values are removed. After preprocessing, FSS (Feature Subset Selection) steps are performed where feature reduction is performed. It also reduced cost of securing the data. After that, information Gain Ratio is performed to rank the target dataset. The advantage of this research is to identify interesting patterns for prediction of churner's behaviour. The disadvantage of this research is that if dataset will be increased then the process will become slow and requires more time for prediction.

*Bharat (2019)* proposed a model based on the activity pattern of customers. Specifically, its measurement is based on the customer's activity by finding the average length of inactive time and frequencies of inactivity. The proposed method can be used in other domains for churn prediction.

*Gajowniczek, Orłowski & Ząbkowski (2019)* used Artificial Neural Network with entropy cost functions for prediction model of customers. Numerical method like classifications tree or SVM provides higher accuracy in classification, which shows the simplest way to apply the new q-error functions to conclude the issue. *Zhang et al. (2018)* stated that customer churn is valuable for telecom companies to retain weighty users. A customer ccp model having more accuracy is very weighty for decision of customer retention. In this paper SVM technique is also used because it is much better for precision. It solves samples under low dimensional space which is linear inspirable in two dimensional space. There is a limitation of proposed model like it is very difficult to quantify churned customers. Therefor there should more complex investigation.

*Ahmed et al. (2020)* proposed a model based on combination of different classifiers in order to create hybrid ensembles model for prediction. In this paper bagged stack learners are proposed. Experimentation is performed on two datasets related to telecom companies. High accuracy is obtained. The benefit of this model is that it does not work on generalized data sets.

*Brownlow et al. (2018)* introduced a new methodology for churn prediction in fund management services and implementation. This framework is based on ensembles learning and a new weighting mechanism is proposed to deal with imbalanced cost sensitivity

problem with financial data. In this model heterogeneous type of data are used collected from different companies. The performance of this model may be increased with extraction and enhancement of learning methods.

*Vo et al. (2018)* used text mining and data mining methods for the prediction of churn. Multiple methods are used for the prediction of churns like semantic information and word importance. This model uses only unstructured data for prediction. In future this research can be extended into segmentation and building personalized recommendation system for different financial services and products.

*Calzada-Infante, Óskarsdóttir & Baesens (2020)* performed comparison of two techniques Time-Order-Graph and Aggregated-Static-Graph with forest classifier using three threshold measures to evaluate the predictive performance of the similarity forest classifier with each centrality metric.

*Nguyen & Duong (2021)* discussed the comparison of two prediction techniques which are SMOTE and Deep Belief Network (DBN) against two cost sensitive learning approaches *e.g.* Focal Loss and Weighted Loss. The results show better performance of Focal Loss and Weighted Loss than SMOTE and DBN.

*Vural, Okay & Yildiz (2020)* proposed a new method based on ANN for churn prediction. In this method two layers of ANN are used to predict churn.

*Jain, Khunteta & Srivastava (2020)* discussed overview of different classification algorithms. These algorithms are Multi-Layer-Perception, KNN measure, Fuzzy Cluster and Deep Learning CNN. After comparison of results Deep Learning CNN shows better results as compared to other classification algorithms.

Although a lot of work has been done on churn prediction but still there is room for improvement. There is a need of churn prediction model which has high prediction accuracy. The proposed modal is based on combination of clustering and classification techniques and attained high prediction accuracy. The summary of literature is shown in Table 1.

## PROPOSED METHODOLOGY

The proposed model has increased the churn prediction performance by using a hybrid model where combinations of different clustering and classification ensembles are introduced. Figure 1 shows different modules of proposed model.

### Data acquisition

The first task of proposed model is data acquisition. Two benchmark churn prediction datasets have been used for model evaluation. These datasets are acquired from online data repositories. First churn prediction dataset is collected from GitHub which is freely available online data repository for research. This dataset contains 5,000 instances with 707 churn customers and 4,293 non-churn customers. The second dataset is collected from Bigml platform which is also freely available online repository containing 3,333 instances with 21 attributes having 483 churns and 2,850 non-churn values. This dataset contains the information about customer's concerns behavioural, demographics and revenue information.

**Table 1 State of the art techniques for customer churn prediction.**

| Refs. | Year | Author | Techniques | Accuracy (%) | Precision (%) | Recall (%) | F-Measure (%) |
|---|---|---|---|---|---|---|---|
| (Maldonado et al., 2021) | 2021 | Maldonado, S., Domínguez, G., Olaya, D., & Verbeke, W. | Logit | 56.43 | – | – | – |
| | | | K Nearest Neighbor | 64.07 | – | – | – |
| | | | CART | 57.05 | – | – | – |
| | | | Random Forest | 66.24 | – | – | – |
| (Adhikary & Gupta, 2020) | 2020 | Adhikary, D. D., & Gupta, D. | Voting | 71.3 | NA | 71.4 | NA |
| | | | Bagging | 71.9 | 69.3 | 71.9 | 62.1 |
| | | | AdaBoost | 71.3 | NA | 71.4 | NA |
| | | | Stacking | 71.3 | NA | 71.4 | NA |
| (Vural, Okay & Yildiz, 2020) | 2020 | Vural, U., Okay, M. E., & Yildiz, E. M | Artificial Neural Network | 89 | – | – | – |
| (Saghir et al., 2019) | 2019 | Saghir, M., Bibi, Z., Bashir, S., & Khan, F. H. | Bagging | 80.8 | 81.88 | 75.28 | 78.44 |
| | | | AdaBoost | 73.9 | 70.46 | 73.74 | 72.06 |
| (Singh & Sivasankar, 2019) | 2019 | Singh, B. E. R., & Sivasankar, E. | Bagging | 79.13 | NA | NA | NA |
| | | | Boosting | 82.03 | NA | NA | NA |
| (Pamina et al., 2019) | 2019 | Pamina, J., Raja, B., SathyaBama | XG | 79.8 | – | – | 58.2 |
| | | | K nearest neighbor | 75.4 | – | – | 49 |
| | | | Random Forest | 77 | – | – | 50.6 |
| (Halibas et al., 2019) | 2019 | Halibas, A. S., Matthew | Gradient Boosted Tree | 79.1 | 73.1 | 79.6 | 76.2 |
| (Amin et al., 2019b) | 2019 | Amin, A., Shah, B., Abbas, | Genetic Algorithm +Naïve Bayes | 89.1 | 95.65 | 16.92 | 28.76 |
| (Saghir et al., 2019) | 2019 | ] Saghir, M., Bibi, Z., Bashir | Ensemble Classifier with Bagging and Neural network | 81 | 81.56 | 73.74 | 72.06 |
| (Ullah et al., 2019) | 2019 | Ullah, I., Raza, B., Malik, A. K | Random Forest | 88 | 89.1 | 89.6 | 87.6 |
| | | | Random tree | 0.85 | – | – | 21.5 |
| (Bharat, 2019) | 2019 | Bharat, A | Logistic Regression | 70 | – | – | – |
| (Gajowniczek, Orłowski & Ząbkowski, 2019) | 2019 | Gajowniczek, K., Orłowski, A. | Entropy Cost Function Neural network | 60 | 74 | 77 | N/A |
| (Vijaya & Sivasankar, 2018) | 2018 | J. Vijaya and E. Sivasankar | K-means | 87.61 | 93.68 | 12.23 | - |
| | | | K-mediods | 90.91 | 98 | 28.4 | – |
| | | | Naïve Bayes | 25.5 | 100 | – | – |
| | | | K Nearest Neighbor | 91.39 | 99 | 01 | – |
| (Höppner et al., 2017) | 2018 | Höppner, S., Stripling, E | EMPC with Decision Tree | 89 | 94.81 | – | 60.7 |
| (Ali et al., 2018) | 2018 | Ali, M., Rehman, A. | Support Vector Machine | 90 | 98.2 | N/A | 98.1 |
| | | | Bagging Stacking | 85.5 | 73.1 | | 78.8 |
| | | | Naïve Bayes | 92.9 | 92.7 | | 92.7 |
| (Amin et al., 2019c) | 2018 | Amin, A., Shah, B., Khattak | Naïve Bayes | 86 | N/A | N/A | 16.7 |
| | | | K Nearest Neighbor | 85 | – | – | 16.6 |
| | | | Gradient Boosted Tree | 72 | – | – | 17.3 |
| | | | SRI | 16.7 | – | – | 87 |
| | | | DP | 80 | – | – | 16.0 |
| (Amin et al., 2018) | 2018 | Amin, A., Shah, B | JIT | 59 | – | – | – |
| (Chen, 2017) | 2018 | Runsha Dong(&), Fei Su, | Support Vector Machine | 70.6 | – | – | – |

| Refs. | Year | Author | Techniques | Accuracy (%) | Precision (%) | Recall (%) | F-Measure (%) |
|---|---|---|---|---|---|---|---|
| (*Vo et al., 2018*) | 2018 | Vo, N. N., Liu, S., Brownlo | XGBoostAlgorithm | 81.08 | – | – | – |
| (*Zhang et al., 2018*) | 2018 | Zhang, X., Zhang, Z., Liang | Decision Tree | 70 | – | – | – |
| (*Zhu et al., 2018*) | 2018 | Zhu, B., Xie, G., Yuan, Y | CART | 67.97 | – | – | 100 |
| (*De Caigny, Coussement & De Bock, 2018*) | 2018 | De Caigny, A., Coussement, K | Logistic regression | 88.12 | – | – | – |
| | | | Decision Tree | 88 | – | – | – |
| (*Amin et al., 2019a*) | 2017 | Amin, A., Al-Obeidat | Naïve Bayes | 57 | 54.14 | 61.30 | 57.50 |
| (*Amin et al., 2018*) | 2017 | Amin, A., Al-Obeidat, F., Shah | JIT | 55.3 | 57.62 | 40.05 | 47.26 |
| (*Stripling et al., 2018*) | 2017 | Stripling, E., vanden Broucke, S | Proflogit | 70 | – | – | – |
| (*Zhu et al., 2018*) | 2017 | Zhu, B., Baesens, B., Backiel, A | Sampling Method | 86.06 | – | – | – |
| (*Mahajan & Som, 2016*) | 2017 | Mahajan, R., & Som, S | K-Local Maximum Features Extraction Method | 80 | 50 | 22 | 94 |
| (*Mishra & Reddy, 2017*) | 2017 | Mishra, A., & Reddy | Bagging | 90.83 | – | 92.02 | – |
| | | | Boosting | 90.32 | | 97.91 | – |
| | | | Random Forest | 91.67 | 83.11 | 98.89 | – |
| | | | Decision Tree | 90.96 | – | – | – |
| (*Tiwari, Sam & Shaikh, 2017*) | 2016 | Tiwari, A., Sam, R., & Shaikh, | Naïve Bayes | 70 | – | – | – |
| (*Petkovski et al., 2016*) | 2016 | Petkovski, A. J., Stojkoska | Naïve Bayes | 85.24 | – | | 82 |
| | | | C4.5 | 91.57 | | | 84 |
| | | | K Nearest Neighbor | 90.59 | | | 85 |
| (*Ahmed & Maheswari, 2017*) | 2016 | Ahmed, A. A., & Maheswari | Firefly Algorithm | 86.38 | 90 | 80 | 93 |
| (*Yu et al., 2018*) | 2016 | Yu, R., An, X., Jin, B., Shi, J., | Particle Classification Optimization Based BP | 69.64 | 87.84 | 51.43 | 48.57 |

## Data preprocessing

The main purpose of data pre-processing is to remove noise, anomalies, missing values and duplication from data (*Azeem, Usman & Fong, 2017*). In pre-processing, a model needs to remove missing values, noisy data, duplication and only needs to use important features from data (*Omar et al., 2021*). Data preprocessing is the first step which is applied on churn prediction data. The proposed churn prediction model has incorporated following tasks during data pre-processing.

- **Data Cleaning:** Prediction is very difficult when there are missing values, duplication and noise in the data. So data cleaning is performed to replace missing values with actual values which are calculated by each attribute mean, remove duplicated data and noise/error values are identified and removed.
- **Feature Selection:** Feature selection is the most important step of data pre-processing. Feature selection is performed using forward selection and most important features are chosen for prediction model.
- **Data Reduction:** In this step data is reduced in smaller volume for producing compact and understandable results.

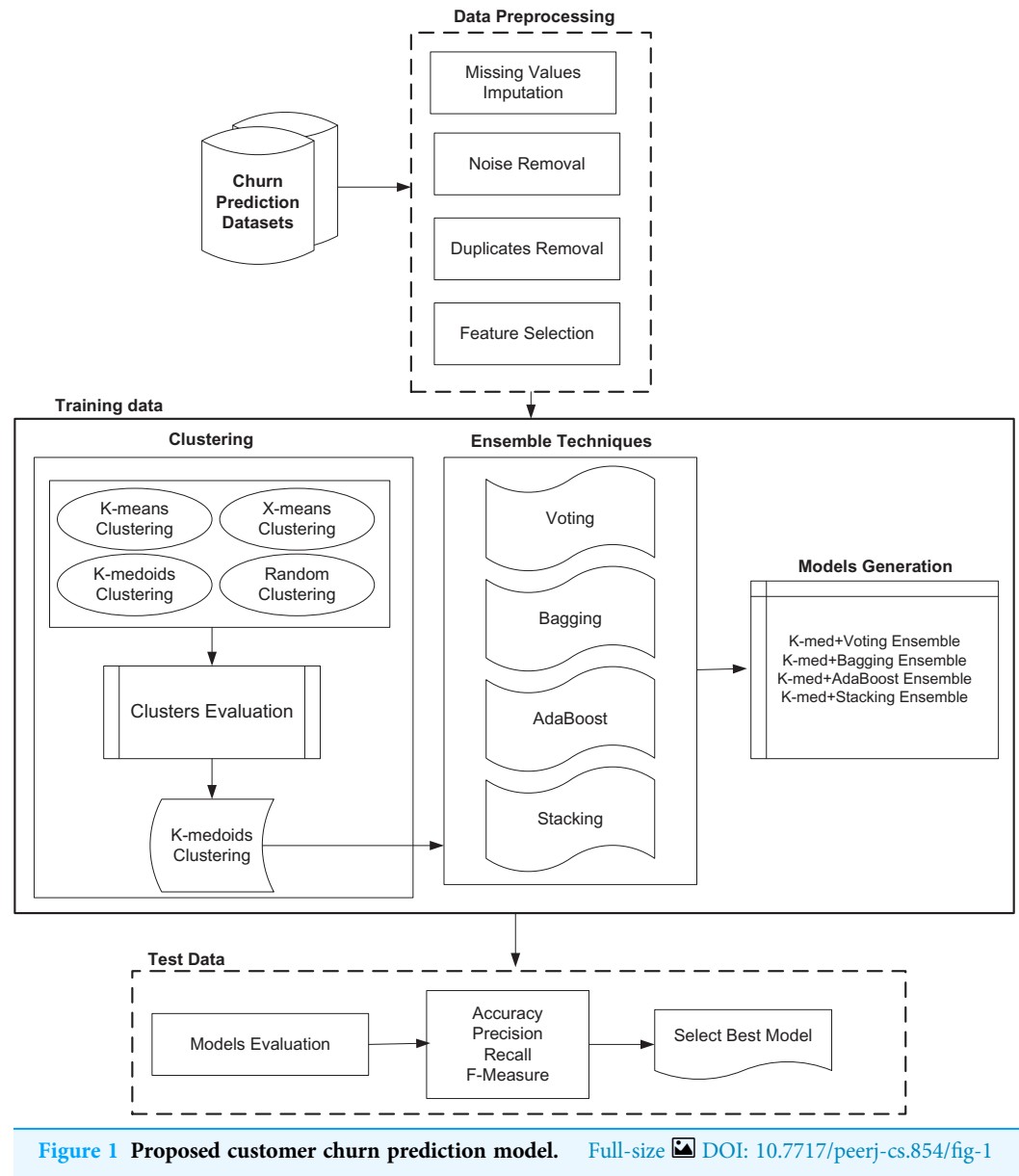

**Figure 1 Proposed customer churn prediction model.**

## Clustering algorithms

After data preprocessing, clustering is applied on cleaned and refined data. The proposed model has employed clustering in order to improve the prediction performance. Following clustering methods have been used by the proposed model.

### K-means clustering

K Means clustering algorithm divides N rows into K segments, and K is always less than N. It randomly selects the value of k which represents centre of cluster mean. It measures the distance between the clusters and compute mean for every cluster. The process continuous iteratively until desired clusters is refined. Following formula is used to measure the distance (*Gajowniczek, Orłowski & Ząbkowski, 2019*).

$$J(V) = \sum_{i=1}^{c} \sum_{j=1}^{c_i} (||x_i - v_j||)^2 \tag{1}$$

where $||x_i - v_j||$ is the Euclidean distance between two clusters.

### K-medoids clustering

In 1987, Rousseeuw Lloyd and Kaufman introduced a clustering technique which is also partitioned based and is termed as K-Medoids algorithm. K-Medoids is more robust to noise and outliers as compared to k-means (*Gajowniczek, Orłowski & Ząbkowski, 2019*). Following formula is used to calculate the cost of each cluster.

$$c = \sum_{c_i} \sum_{p_i \in c_i} |P_i - C_i| \tag{2}$$

where $P_i$ and $C_i$ are objects for which dissimilarity is calculated.

### X-means clustering

X means clustering is a variation of k means clustering where clusters are refined and subdivided repeatedly until Bayesian Information Criteria (BIC) is reached. The efficient estimation of number of clusters is obtained automatically instead of take input from user in the form of K. Covariance of each cluster is measured and following formula is used to calculate the variance (*Gajowniczek, Orłowski & Ząbkowski, 2019*).

$$\sigma^2 = \frac{1}{R - K} \sum_{i} (x_i - \mu_{(i)})^2 \tag{3}$$

where $R$ and $K$ are number of points and number of clusters respectively and μ is the centroid of i cluster.

### Random clustering

Random clustering is often used in Rapidminer to perform random flat clustering of given dataset. Moreover, some of the clusters can be empty and the samples are assigned to clusters randomly.

## Classification and prediction algorithms

After performing clustering, Classification is performed by the proposed model. Each clustering method is evaluated and best clustering method is combined with classification algorithms. The proposed model first used single classifiers and their performance is measures. Then, ensemble classifiers are used along with clustering to attain the highest prediction accuracy. The combination of clustering and ensemble classifier which has attained highest churn prediction accuracy will be considered as proposed model. Following classifiers are used by the proposed churn prediction model.

### K-nearest neighbor

The k-nearest neighbor is one of the simplest classification methods in data mining. Following distance formula is used to measure the distance (*Gajowniczek, Orłowski & Ząbkowski, 2019*):

$$d(x,y) = \sqrt{\sum_{i=1}^{k}(x_i - y_i)^2} \tag{4}$$

where $k$ is number of samples in training data $x$ and $y$ are instances for which distance is calculated.

### Decision tree

Quinlan introduced in 1993 a divide and conquer method and termed as decision tree. Entropy and then information gain is calculated for each attribute using the following formulas.

$$H(S) = \sum_{x \in X} -p(x)\log_2 p(x) \tag{5}$$

where $S$ is the dataset, $X$ is set of classes in $S$ and $p(x)$ is probability of each class.

$$IG(S, A) = H(S) - \sum_{t \in T} p(t)H(t) = H(S) - H(S|A) \tag{6}$$

where $H(S)$ is entropy of S, $T$ is the subset of $s$, $p(t)$ is probability of subset $t$ and $H(t)$ is entropy of subset $t$.

### Gradient boosted tree

The idea behind GBT is to improve the prediction accuracy by producing ensemble of decision trees. GBT outperforms random forest as it produces the ensemble of weak prediction models. The prediction is given as follows:

$$\hat{y_i} = \sum_{j} \theta_j x_{ij} \tag{7}$$

where $y$ is the prediction made by input $x$. $\Theta$ is the best parameter that best fits the model.

### Random forest

Random forest or random decision forest is an ensemble learning method used for classification, regression and other tasks that operate by constructing a multitude of decision tree at training time and outputting the class that is the mode of the classes or average prediction (regression) of the individual trees. Entropy or Gini index are used for tree construction using following formulas.

$$Gini = 1 - \sum_{i=1}^{c}(p_i)^2 \tag{8}$$

$$Entropy = \sum_{i=1}^{c} -p_i * \log_2 p_i \tag{9}$$

### Deep learning

It is a multi-layer technique which compares large number of layers of neurons. It is an artificial network which is used to solve more complex and difficult problems of data mining.

### Naïve Bayes

Naive Bayes method is a supervised learning algorithm based on applying Bayes' theorem. Following formula is used by the Naïve Bayes classifier.

$$P(c|x) = \frac{P(x|c)P(c)}{P(x)} \tag{10}$$

where $P(c)$ is the prior probability of class, $P(x)$ is prior probability of predictor, $p(x|c)$ is probability of predictor given class and $p(x|c)$ is posterior probability of class.

### NB (K) (Naïve Bayes Kernel)

The Naive Bayes (Kernel) operator can be applied on numerical attributes. A kernel is a weighting function used in non-parametric estimation techniques. Kernels are used in kernel density estimation to estimate random variables' density functions, or in kernel regression to estimate the conditional expectation of a random variable.

### Ensemble classifiers

*Krawczyk et al. (2017)* used ensemble methods to apply multiple learning algorithms for prediction. Ensembles increase the performance of the system or model (*Rustam et al., 2020*).

Following ensemble models are used by the proposed model.

#### Voting

Voting method is used to combine the results of individual classification algorithms using majority voting. Each individual classifier assigns a class label to test data, then their results are combined using voting and final class prediction is generated using maximum number of votes for a particular class (*Gupta & Chandra, 2020*). Following formula is used to apply majority voting on dataset:

$$\sum_{t=1}^{T} d_{t,J}(x) = max_{j=1,2,3...C} \sum_{t=1}^{T} d_{t,J}| \tag{11}$$

where $T$ represents the number of classifiers, and $d(t,J)$ is the decision of classifier and $J$ represents the classes.

#### Bagging

Bagging stands for Bootstrap Aggregation. It is an ensemble classifier which has bag of similar and dissimilar objects. It helps to decrease the variance of the classifiers which are used in prediction model to make better performance (*Brown et al., 2005*). Then evaluation of Bagging is given as follows:

$$V_{t,j} = \begin{cases} 1 & \text{if } h_t \text{ picks class } w_j \\ 0 & \text{otherwise} \end{cases} \tag{12}$$

where $t$ represents training samples, $h_t$ represents trained classifiers and $w_i$ represents class labels. Each class will have total votes represented by:

$$V_j = \sum_{t=1}^{T} v_{t,j} \quad j = 1, 2, 3 \ldots c \tag{13}$$

*AdaBoost*

AdaBoost is the short form of Adaptive Boosting, is a Meta algorithm, which can be used in conjunction with different other learning algorithms to improve their performance (*Brown et al., 2005*). Weighted majority voting is applied on the classifiers. Every classifier gets equal opportunity to draw samples in each iteration. Following formula is used to apply weighted majority voting:

$$V_j = \sum_{t:h_t(x)=w_j} \log\left(\frac{1}{\beta_t}\right) \quad j = 1, 2, 3 \ldots C \tag{14}$$

where $\beta_t$ is normalized error, $h_t$ represents trained classifiers and $w_j$ represents class labels of training data.

*Stacking*

Stacking is used for combining different leaners rather than selecting among them. It can be used for getting a performance better than any single one of the trained models. Bootstrapped samples of training data are used to train the classifiers. There are two types of classifiers used in stacking, Tier-1 classifiers and Tier-2 classifiers. Tier-1 classifiers are trained on bootstrapped samples and generate prediction, their result are then used to train Tier-2 classifiers. This way training data is properly used to perform learning (*Brown et al., 2005*).

## Working of proposed model

The proposed model has increased the churn prediction performance by using a hybrid model where clustering methods and classification methods are combined. Combinations of different clustering and classification ensembles are introduced and best combination models are selected for final prediction. First clustering is used to generate the clusters of given dataset. "Map clustering on Label" operator is used to assign labels to data. Then classification is performed for labelled data to generate the results.

It is also proved that performance; accuracy and efficiency of churn prediction model can be increased by using the proposed novel hybrid models. As single classifier based model cannot provide high accuracy, therefore the proposed models used the hybrid model for prediction of churn.

1. First of all clustering evaluation is carried out and results are obtained and select best clustering technique on the behalf of accuracy.
2. After clustering, single classifier based classification is performed and then accuracy, precision, recall and f-measure results are obtained.
3. After that hybrid model of best clustering method and each single classifier is developed and performance results are obtained for each dataset.

4. Next, only ensemble classifiers based models are developed and evaluated on both datasets.

5. Then, these ensemble models are combined with best clustering technique in order to make hybrid models and performance is evaluated. It is clear from the evaluation that proposed combination of clustering and ensemble models has achieved highest prediction accuracy as compared to state of the art models for both churn prediction datasets.

## EXPERIMENTAL RESULTS, EVALUATION AND DISCUSSION

The experimental of proposed model is performed on two benchmark churn prediction datasets. First, clustering techniques are performed on each dataset and best clustering method is selected. Next, single classifiers are performed on each dataset and their performance is evaluated, shown in "Map Clustering on Label". Then, single classifiers are evaluated along with K-Medoid clustering and again their performance is evaluated for each dataset as shown in "Clustering with Single Classifier on Churn Prediction Datasets". It is analyzed that performance of single classifiers is improved. Afterthat, ensembles (Voting, Bagging, Stacking, AdaBoost) are evaluated on each dataset along with K-Medoid clustering as shown in "Clustering with Ensembles on Churn Prediction Datasets". The analysis indicates that AdaBoost ensemble along with clustering performed better as compared to other ensembles for both churn prediction datasets.

Following datasets are used for the experiments and evaluation which are freely available at online data repositories.

### GitHub dataset

First churn prediction dataset is collected from GitHub which is an online data repository. The datasets are freely available over here for research. The dataset name is "Kaggle-telecom-customer-churn-prediction" obtained from the data source https://www.kaggle.com/blastchar/telco-customer-churn. It is used to predict customer's behaviour to retain them. It contains 5,000 instances data where each row represents a customer and columns represents customer's attributes. The dataset contains 707 churn customers and 4,293 non-churn customers.

### BigML dataset

The second dataset is collected from Bigml platform which is also freely available online data repository. The dataset is obtained from data source https://cleverdata.io/en/bigdata-predictions-bigml/. The name of dataset is "Churn in Telecom's dataset". It contains 3,333 instances having 21 attributes. There are 483 churns and 2,850 non churn customers in the dataset. This dataset is also used to predict the customer's behaviour.

### Rapid miner

Rapid Miner is a data science software platform that provides an integrated environment for data preparation, machine learning, deep learning, text mining, and predictive

analytics. The proposed research is implemented using Rapid Miner. It is also freely available on the web.

## Model evaluation

Confusion matric is used for model evaluation. With the help of confusion matrix performance of proposed model is analysed. The performance of proposed model is analysed using accuracy, precision, recall and f-measure (*Rupapara et al., 2021*; *Jamil et al., 2021*; *Rustam et al., 2021*). These parameters can be measured with the help of following formulae where TP represents true positives, TN is true negatives, FP shows false positives and FN shows false negatives.

$$Accuracy = \frac{TP + TN}{TP + TN + FP + FN} \tag{15}$$

$$Precision = \frac{TP}{TP + FP} \tag{16}$$

$$Recall = \frac{TP}{TP + FN} \tag{17}$$

$$F{-}Measure = \frac{2 * Precision * Recall}{Precision + Recall} \tag{18}$$

## Map clustering on label

After clustering, mapping is used to generate TP, TN, FN, and FP from datasets. Mapping maps the cluster 0 and cluster 1 with churner and non-churner *i.e.* 0 and 1. For example we have a table containing three columns cluster0, cluster 1 and churn; mapping generates another column prediction (churn) which contains class 0 and class 1. It maps the values of cluster 0 and clusters 1 with class 0 and class 1 and then TP, TN, FN, and FP can be generated.

After applying clustering technique on dataset, there are two clusters which are cluster_0 and cluster_1. These clusters are mapped with prediction class and check whether the values of class lie in clusters or not. So for this purpose clusters are mapped with Prediction (Class) and analysed. If values of cluster_0 lie in prediction class 0, it generates TN, and if values of cluster_1 lie in Prediction (Class) 1 it generates TP. Similarly, if values of cluster_0 lies in Prediction (class) 1 it generates FP, and if values of cluster_1 lies in Prediction (Class) 0 it generates FN. Therefore, cluster evaluation is performed with the help of mapping. Table 2 shows clustering evaluation results on GITHUB and Bigml datasets.

## Clustering with single classifier on churn prediction datasets

As it is clear from the literature that single classification techniques show low classification accuracy as compared to hybrid model, therefore now supervised and unsupervised techniques are combined to generate hybrid model and then this hybrid model will be used for classification in order to increase the accuracy level. It is analysed from Table 2 that k-med shows higher accuracy as compared to other clustering techniques therefore now

**Table 2 Clustering evaluation on churn prediction datasets.**

| Technique | Accuracy | Recall | Precision | F-measure | RMSE | MSE | MAE |
|---|---|---|---|---|---|---|---|
| **Clustering evaluation on GitHub churn prediction dataset** | | | | | | | |
| X-means | 50.58 | 52.05 | 14.72 | 22.94 | 0.78 | 0.6084 | 0.15 |
| K-means | 50.58 | 52.05 | 14.72 | 22.94 | 0.78 | 0.6084 | 0.15 |
| K-med | 65.44 | 29.13 | 14.37 | 19.25 | 0.69 | 0.4761 | 0.11 |
| Random | 50.96 | 48.93 | 14.19 | 22.01 | 0.75 | 0.5625 | 0.14 |
| **Clustering Evaluation on Bigml churn prediction dataset** | | | | | | | |
| X-means | 50.04 | 48.86 | 14.26 | 22.08 | 0.63 | 0.3969 | 0.0992 |
| K-means | 50.04 | 48.86 | 14.26 | 22.08 | 0.63 | 0.3969 | 0.0992 |
| K-med | 55.56 | 41.82 | 14.40 | 21.43 | 0.54 | 0.2916 | 0.0729 |
| Random | 50.94 | 49.06 | 14.57 | 22.47 | 0.61 | 0.3721 | 0.093 |

**Table 3 Clustering with single classifier on churn prediction datasets.**

| Technique | GitHub dataset | | | | Bigml dataset | | | |
|---|---|---|---|---|---|---|---|---|
| | Accuracy | Recall | Precision | F-measure | Accuracy | Recall | Precision | F-measure |
| K med+GBT | 94 | 62.23 | 93.02 | 74.57 | 92.25 | 51.96 | 90.61 | 66.05 |
| K med+DT | 86.4 | 4.10 | 93.54 | 7.85 | 86.76 | 8.90 | 97.72 | 16.31 |
| K med+RF | 87.6 | 14.14 | 88.49 | 24.39 | 91.53 | 65.42 | 73.31 | 69.14 |
| K med+KNN | 86.02 | 12.58 | 52.35 | 20.29 | 84.39 | 2.61 | 20.63 | 4.76 |
| K med+DL | 92.5 | 68.45 | 76.10 | 72.07 | 91.53 | 65.42 | 73.31 | 69.14 |
| K med+NB | 87.02 | 52.61 | 54.22 | 53.40 | 83.28 | 43.47 | 42.51 | 42.98 |
| K med+NB(K) | 83.56 | 42.43 | 41.95 | 42.19 | 91.14 | 52.79 | 79.19 | 63.35 |
| Average | 88.15 | 36.65 | 71.38 | 42.11 | 88.70 | 41.52 | 68.18 | 47.39 |

the combination of k-med with seven different classification algorithms (GBT, DT, RF, kNN, DL, NB, NB(K)) is applied one by one on each dataset as shown in Table 3.

## Clustering with ensembles on churn prediction datasets

Tables 4–7 show that k-med clustering is combined with different combination of classifiers. Voting, Bagging, Stacking and AdaBoost ensembles are used. The combination of GBT, DT and DL shows highest accuracy when it is combined with k-med clustering.

## Comparison of different techniques

Now the comparison of all techniques has been carried out. The comparison shows different levels of accuracy for different hybrid models. The average accuracy of different techniques has been compared. Table 8 shows the comparison results. As it is clear from experiments that results are improved on each step because a hybrid approach is used to improve the results.

## Comparison with state of the art techniques

Tables 9 and 10 show the comparison of proposed model with different state of the art techniques. Proposed model shows higher accuracy as compared to existing techniques.

Table 4 Clustering with voting ensemble for churn prediction datasets.

| Technique | GitHub dataset | | | | Bigml dataset | | | |
|---|---|---|---|---|---|---|---|---|
| | Accuracy | Recall | Precision | F-measure | Accuracy | Recall | Precision | F-measure |
| K-med+GBT+DT+RF | 89.06 | 24.04 | 94.44 | 38.33 | 87.51 | 14.28 | 97.18 | 24.90 |
| K-med+GBT+DT+KNN | 89.62 | 28.14 | 94.76 | 43.40 | 86.04 | 4.14 | 90.90 | 7.92 |
| K-med+GBT+DT+DL | 94.06 | 61.52 | 94.56 | 74.55 | 92.40 | 51.34 | 93.23 | 66.22 |
| K-med+GBT+DT+NB | 92.58 | 53.18 | 90.38 | 66.96 | 91.92 | 45.34 | 97.76 | 61.95 |
| K-med+GBT+DT+NB(K) | 91.3 | 44.97 | 87.36 | 59.38 | 89.46 | 32.09 | 87.07 | 46.89 |
| K-med+DT+RF+KNN | 88.14 | 16.54 | 97.5 | 28.29 | 86.01 | 3.72 | 94.73 | 7.17 |
| K-med+DT+RF+DL | 88.94 | 23.62 | 92.77 | 37.65 | 87.78 | 15.94 | 98.71 | 27.45 |
| K-med+DT+RF+NB | 88.48 | 20.79 | 90.18 | 33.79 | 87.57 | 14.90 | 96 | 25.80 |
| K-med+DT+RF+NB(K) | 88.44 | 19.66 | 93.28 | 32.47 | 87.30 | 13.45 | 92.85 | 23.50 |
| K-med+RF+KNN+DL | 88.42 | 19.80 | 92.10 | 32.59 | 87.37 | 14.83 | 96.05 | 25.70 |
| K-med+RF+KNN+NB | 88.24 | 18.59 | 91.60 | 30.82 | 87.42 | 14.69 | 91.02 | 25.31 |
| K-med+RF+KNN+NB(K) | 88.14 | 20.36 | 82.75 | 32.69 | 87.18 | 12.62 | 92.42 | 22.22 |
| K-med+NB+NB(K)+KNN | 88.42 | 34.79 | 67.58 | 45.93 | 88.44 | 32.71 | 72.47 | 45.07 |
| K-med+NB+NB(K)+DL | 90.36 | 54.87 | 70.41 | 61.68 | 89.85 | 55.69 | 68.44 | 61.41 |
| Average | 89.58 | 31.49 | 88.55 | 44.18 | 88.31 | 23.27 | 90.63 | 33.68 |

Table 5 Clustering with bagging ensemble on churn prediction datasets.

| Technique | GitHub dataset | | | | Bigml dataset | | | |
|---|---|---|---|---|---|---|---|---|
| | Accuracy | Recall | Precision | F-measure | Accuracy | Recall | Precision | F-measure |
| K-med+GBT+DT+RF | 89.12 | 24.46 | 94.53 | 38.87 | 87.54 | 14.49 | 97.22 | 25.22 |
| K-med+GBT+DT+kNN | 89.7 | 28.71 | 94.85 | 44.08 | 86.10 | 4.55 | 91.66 | 8.67 |
| K-med+GBT+DT+DL | 94.12 | 61.10 | 95.78 | 74.61 | 92.41 | 51.55 | 93.25 | 66.4 |
| K-med+GBT+DT+NB | 92.64 | 53.60 | 90.45 | 67.31 | 91.98 | 45.75 | 97.78 | 62.34 |
| K-med+GBT+DT+NB(K) | 91.4 | 45.68 | 87.53 | 60.03 | 89.52 | 32.50 | 87.22 | 47.36 |
| K-med+DT+RF+kNN | 88.1 | 16.26 | 97.45 | 27.87 | 86.04 | 3.93 | 95 | 7.55 |
| K-med+DT+RF+DL | 88.98 | 23.90 | 92.85 | 38.02 | 87.81 | 16.14 | 98.73 | 27.75 |
| K-med+DT+RF+NB | 88.5 | 20.79 | 90.74 | 33.83 | 87.66 | 15.52 | 96.15 | 26.73 |
| K-med+DT+RF+NB(K) | 88.42 | 19.23 | 94.44 | 31.96 | 87.33 | 13.66 | 92.95 | 23.82 |
| K-med+RF+kNN+DL | 88.32 | 18.52 | 94.24 | 30.96 | 87.63 | 15.32 | 96.10 | 26.42 |
| K-med+RF+kNN+NB | 88.04 | 16.83 | 92.24 | 28.46 | 87.45 | 14.90 | 91.13 | 25.62 |
| K-med+RF+kNN+NB(K) | 88 | 17.11 | 89.62 | 28.74 | 87.24 | 13.04 | 92.64 | 22.86 |
| K-med+NB+NB(K)+kNN | 88.66 | 33.94 | 70.58 | 45.84 | 88.47 | 32.91 | 72.60 | 45.29 |
| K-med+NB+NB(K)+DL | 90.62 | 54.87 | 72.11 | 62.32 | 89.88 | 55.90 | 68.52 | 61.57 |
| Average | 89.61 | 31.07 | 89.82 | 43.78 | 88.37 | 23.58 | 90.78 | 34.11 |

In this research hybrid models of supervised and unsupervised learning techniques is proposed and implemented with rapid miner. These models are applied on two datasets which are freely available on online data repositories. In first step clustering algorithms are selected *i.e.* k-means, K-medoid, X-means and Random Clustering are selected for

**Table 6 Clustering with stacking ensemble on churn prediction datasets.**

| | GitHub dataset | | | | Bigml dataset | | | |
|---|---|---|---|---|---|---|---|---|
| Technique | Accuracy | Recall | Precision | F-measure | Accuracy | Recall | Precision | F-measure |
| K-med+GBT+DT+RF | 94.34 | 76.09 | 82.51 | 79.17 | 92.39 | 67.49 | 77.80 | 72.28 |
| K-med+GBT+DT+KNN | 94.56 | 73.26 | 86.18 | 79.20 | 92.31 | 66.04 | 77.61 | 71.36 |
| K-med+GBT+DT+DL | 94.65 | 73.12 | 87.33 | 79.59 | 92.40 | 66.25 | 78.04 | 71.66 |
| K-med+GBT+DT+NB | 94.5 | 74.82 | 84.50 | 79.36 | 89.97 | 72.04 | 63.61 | 67.57 |
| K-med+GBT+DT+NB(K) | 94.5 | 76.23 | 83.43 | 79.67 | 91.47 | 66.66 | 72.35 | 69.39 |
| K-med+DT+RF+KNN | 86.02 | 12.58 | 52.35 | 20.29 | 87.12 | 13.66 | 84.61 | 23.52 |
| K-med+DT+RF+DL | 92.38 | 69.02 | 75.07 | 71.92 | 91.32 | 68.73 | 70.63 | 69.67 |
| K-med+DT+RF+NB | 86.86 | 49.50 | 53.84 | 51.58 | 91.50 | 44.72 | 93.10 | 60.41 |
| K-med+DT+RF+NB(K) | 87.72 | 14.56 | 91.15 | 25.12 | 87.57 | 16.14 | 89.65 | 27.36 |
| K-med+RF+KNN+DL | 92.2 | 67.04 | 75.11 | 70.85 | 91.05 | 64.18 | 71.26 | 67.53 |
| K-med+RF+KNN+NB | 86.02 | 12.58 | 52.35 | 20.29 | 86.28 | 50.51 | 52.81 | 51.64 |
| K-med+RF+KNN+NB(K) | 86.36 | 49.08 | 51.86 | 50.43 | 87.21 | 14.07 | 86.07 | 24.19 |
| K-med+NB+NB(K)+KNN | 85.94 | 51.06 | 50.27 | 50.66 | 90.63 | 62.52 | 69.74 | 65.93 |
| K-med+NB+NB(K)+DL | 92.96 | 69.73 | 78.12 | 73.69 | 86.82 | 55.48 | 54.47 | 54.97 |
| Average | 90.66 | 55.05 | 71.70 | 59.49 | 89.87 | 52.05 | 74.42 | 56.97 |

**Table 7 Clustering with stacking ensemble on churn prediction datasets.**

| | GitHub dataset | | | | Bigml dataset | | | |
|---|---|---|---|---|---|---|---|---|
| Technique | Accuracy | Recall | Precision | F-measure | Accuracy | Recall | Precision | F-measure |
| K-med+GBT+DT+RF | 87.7 | 13.86 | 94.23 | 24.16 | 88.83 | 25.46 | 91.11 | 39.80 |
| K-med+GBT+DT+KNN | 87.74 | 13.57 | 97.95 | 23.85 | 87.69 | 19.04 | 82.88 | 30.97 |
| K-med+GBT+DT+DL | 94.7 | 75.10 | 87.04 | 80.63 | 92.43 | 66.45 | 78.10 | 71.81 |
| K-med+GBT+DT+NB | 94.48 | 72.70 | 86.09 | 78.83 | 90.03 | 51.13 | 72.01 | 59.80 |
| K-med+GBT+DT+NB(K) | 93 | 59.97 | 86.35 | 70.78 | 88.23 | 39.95 | 65.42 | 49.61 |
| K-med+DT+RF+KNN | 87.02 | 8.48 | 96.77 | 15.60 | 87.57 | 15.52 | 92.59 | 26.59 |
| K-med+DT+RF+DL | 87.72 | 13.57 | 96.96 | 23.82 | 89.07 | 27.32 | 91.03 | 42.03 |
| K-med+DT+RF+NB | 87.64 | 13.71 | 92.38 | 23.89 | 88.32 | 24.63 | 82.63 | 37.95 |
| K-med+DT+RF+NB(K) | 87.52 | 13.29 | 89.52 | 23.15 | 88.17 | 19.95 | 91.42 | 32.76 |
| K-med+RF+KNN+DL | 88.22 | 19.09 | 88.81 | 31.43 | 87.75 | 16.56 | 94.11 | 28.16 |
| K-med+RF+KNN+NB | 88.08 | 17.96 | 88.81 | 29.88 | 87.54 | 15.52 | 91.46 | 26.54 |
| K-med+RF+KNN+NB(K) | 87.7 | 17.68 | 79.11 | 28.90 | 87.30 | 14.28 | 88.46 | 24.59 |
| K-med+NB+NB(K)+KNN | 88.14 | 36.06 | 64.39 | 46.23 | 87.24 | 36.02 | 60 | 45.01 |
| K-med+NB+NB(K)+DL | 90.12 | 58.41 | 67.37 | 62.57 | 89.31 | 55.90 | 65.37 | 60.26 |
| Average | 89.29 | 30.82 | 87.66 | 40.24 | 88.54 | 30.54 | 81.89 | 41.13 |

experimentation. After evaluation of these clustering algorithms it is noticed that the k-medoid showed high accuracy as compered other three clustering algorithms, so K-medoid is selected for hybrid model generation. After selection of clustering algorithm next step is selection of classification algorithm. Seven different classification algorithms

**Table 8 Average accuracy comparison of different techniques.**

| Technique | GitHub dataset | | | |
| --- | --- | --- | --- | --- |
| | Accuracy | Precision | Recall | F-Measure |
| Clustering | 54.39 | 45.54 | 14.50 | 22 |
| Classification(Single Classifier) | 88.14 | 36.75 | 71.24 | 48.49 |
| Clustering with Single Classifier | 88.15 | 36.65 | 71.38 | 42.11 |
| Clustering with Voting Ensemble Classifier | 89.58 | 31.49 | 88.55 | 44.18 |
| Clustering with Bagging Ensemble Classifier | 89.61 | 31.07 | 89.82 | 43.78 |
| Clustering with Stacking Ensemble Classifier | 89.29 | 30.82 | 87.66 | 40.24 |
| Clustering with AdaBoost Ensemble Classifier | 90.66 | 55.05 | 71.70 | 59.49 |
| | Bigml dataset | | | |
| Clustering | 51.65 | 47.15 | 14.38 | 22.03 |
| Classification(Single Classifier) | 87.30 | 33.56 | 65.84 | 44.46 |
| Clustering with Single Classifier | 88.70 | 41.52 | 68.18 | 47.39 |
| Clustering with Voting Ensemble Classifier | 88.31 | 23.27 | 90.63 | 33.68 |
| Clustering with Bagging Ensemble Classifier | 88.37 | 23.58 | 90.78 | 34.11 |
| Clustering with Stacking Ensemble Classifier | 88.54 | 30.54 | 81.89 | 41.13 |
| Clustering with AdaBoost Ensemble Classifier | 89.87 | 52.05 | 74.42 | 56.97 |

**Table 9 Comparison with state of the art techniques with Bigml dataset.**

| Techniques | | References | Accuracy | Precision | Recall | F-Measure | Standard Dev |
| --- | --- | --- | --- | --- | --- | --- | --- |
| Existing models with Bigml dataset | JIT | (Mishra & Reddy, 2017) | 77.27 | 96.02 | 57.25 | 71.42 | NA |
| | UDT | (Amin et al., 2019) | 84.0 | 52.38 | 64.71 | 57.89 | NA |
| | Multilayer Perception | (Yu et al., 2018) | 89.29 | 86.8 | 89.5 | 88.8 | NA |
| | Random Forest | (Tiwari, Sam & Shaikh, 2017) | 89.59 | 89.1 | 89.6 | 87.6 | NA |
| | Bagging + Deep Learning | (Maldonado et al., 2021) | 91.51 | 90.67 | 72.94 | 80.84 | NA |
| | EWD | (Rustam et al., 2020) | 88 | 86.01 | 78 | 79.01 | NA |
| | NB+LR | (Rustam et al., 2021) | 84.51 | 58.18 | 10.92 | 18.39 | NA |
| | Bagging | (Ullah et al., 2019) | 88.3 | 86.8 | 88.3 | 86.4 | NA |
| | AdaBoost | | 86.8 | 84.6 | 86.8 | 84.8 | NA |
| Deep learning models | Long Short Term Memory (LSTM) | | 85.1 | 77.5 | 76.5 | 76.9 | NA |
| | Gated Recurrent Unit (GRU) | | 88.6 | 83.9 | 81.1 | 82.4 | NA |
| | Convolutional Neural Network (CNN) | | 87.9 | 85.6 | 82.7 | 84.12 | NA |
| Proposed models with Bigml dataset | K-med+GBT+DT+DL+Voting | | 92.40 | 93.23 | 51.34 | 66.22 | 0.14 |
| | K-med+GBT+DT+DL+Bagging | | 92.41 | 93.25 | 51.55 | 66.4 | 0.12 |
| | K-med+GBT+DT+DL+Stacking | | 92.40 | 78.04 | 66.25 | 71.66 | 0.15 |
| | K-med+GBT+DT+DL +Adaboost | | 92.43 | 78.10 | 66.45 | 71.81 | 0.10 |

**Table 10 Comparison with state of the art techniques for GitHub dataset.**

| Techniques | | Reference | Accuracy | Precision | Recall | F-Measure | Standard Dev |
|---|---|---|---|---|---|---|---|
| Existing models with Github dataset | FLIC/FDT | (Omar et al., 2021) | 81.5 | – | – | – | NA |
| | K-Means+DT | (Kumar & Kumar, 2019) | 84.26 | 95.68 | – | 90.00 | NA |
| | Bagging + DL | (Maldonado et al., 2021) | 50 | 25 | 50 | 33.33 | NA |
| | AdaBoost+ MLP | | 66.3 | 66.64 | 66.28 | 66.46 | NA |
| | Majority Voting DL+NN+ML | | 66.69 | 67.52 | 66.69 | 67.1 | NA |
| | Bagging+ MLP | | 67.57 | 71.54 | 67.57 | 69.5 | NA |
| | Bagging | (Saghir et al., 2019) | 80.8 | 81.88 | 75.28 | 78.44 | NA |
| | AdaBoost | | 73.9 | 70.46 | 73.74 | 72.06 | NA |
| Deep learning models | Long Short Term Memory (LSTM) | | 90.04 | 84.8 | 79.7 | 82.1 | NA |
| | Gated Recurrent Unit (GRU) | | 90.1 | 85.9 | 83.4 | 84.6 | NA |
| | Convolutional Neural Network (CNN) | | 89.8 | 83.1 | 82.2 | 82.6 | NA |
| Proposed models with GitHub dataset | K-med+GBT+DT+DL+Voting | | 94.06 | 94.56 | 61.52 | 74.55 | 0.14 |
| | K-med+GBT+DT+DL+Bagging | | 94.12 | 95.78 | 61.10 | 74.61 | 0.13 |
| | K-med+GBT+DT+DL+Stacking | | 94.65 | 87.33 | 73.12 | 79.59 | 0.11 |
| | K-med+GBT+DT+DL +Adaboost | | 94.7 | 87.04 | 75.10 | 80.63 | 0.12 |

are selected based on literature which is GBT, DT, RF, KNN, DL, NB, NB (K) for classification. These algorithms have high performance for churn prediction datasets. After evaluation it is noticed that GBT shows high accuracy level. After separate single experimentation a hybrid model of clustering and single classification algorithm is developed to perform the results. In this experimentation K-med and GBT shows better accuracy as compared to other combinations. After combination of single classification algorithm and clustering a hybrid model of classification algorithms is implemented and experiments are performed. The main reason of implementation of hybrid model is that it shows better accuracy as compared to single classifiers (Khairandish et al., 2021; Sujatha et al., 2021), so different combination of above mentioned classifiers are used for experimentation. These hybrid classifiers are used with k-med clustering for better results. After this step ensemble classifiers voting, bagging, adaBoost and stacking are used with hybrid model of clustering and classification. These ensemble classifiers are used to increase the accuracy level. With the combination of ensemble classifiers and clustering algorithm, models show better accuracy for churn prediction.

## CONCLUSION AND FUTURE WORK

Customer churn prediction is a critical problem for telecom companies. The identification of customers that are unhappy with the services provided allows the companies to work

on their weak points, pricing plans, promotions and customer preferences to reduce the reasons for churn. Many techniques are used in literature to predict customer's churn. The proposed research focused on introducing different models for customer churn prediction with high accuracy. Novel hybrid models were introduced by combining clustering and classification approaches. The proposed models were then evaluated on two churn prediction datasets obtained from online data repositories. The analysis of results show that proposed models have achieved higher classification and prediction accuracy as compared to existing state of the art models. In this work the combination of k-med clustering and GBT, DT and DL classifier ensemble shows higher accuracy when compared to other methods.

This research can be extended in future by using big data analytics. Social network analysis can be used to identify the customer's satisfaction level towards telecom services and then these services can be offered to reduce the churn rate. Further datasets can also be used to increase the confidence level on results. Finally, the models can be applied on different sectors like banking, insurance or airline and prediction accuracy can be compared.

### Funding
The authors received no funding for this work.

### Competing Interests
The authors declare that they have no competing interests.

### Author Contributions
- Syed Fakhar Bilal conceived and designed the experiments, performed the experiments, analyzed the data, performed the computation work, prepared figures and/or tables, authored or reviewed drafts of the paper, and approved the final draft.
- Abdulwahab Ali Almazroi conceived and designed the experiments, performed the experiments, analyzed the data, performed the computation work, prepared figures and/or tables, authored or reviewed drafts of the paper, and approved the final draft.
- Saba Bashir conceived and designed the experiments, performed the experiments, analyzed the data, performed the computation work, prepared figures and/or tables, authored or reviewed drafts of the paper, and approved the final draft.
- Farhan Hassan Khan conceived and designed the experiments, performed the experiments, analyzed the data, performed the computation work, prepared figures and/or tables, authored or reviewed drafts of the paper, and approved the final draft.
- Abdulaleem Ali Almazroi conceived and designed the experiments, performed the experiments, analyzed the data, performed the computation work, prepared figures and/or tables, authored or reviewed drafts of the paper, and approved the final draft.

## Data Availability

The data is available at figshare: Bashir, Dr. Saba (2022): archive.zip. figshare. Dataset. https://doi.org/10.6084/m9.figshare.18130610.v1, and at cleverdata: https://cleverdata.io/en/bigdata-predictions-bigml/.

## Supplemental Information

Supplemental information for this article can be found online at http://dx.doi.org/10.7717/peerj-cs.854#supplemental-information.

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
