# Peer review of "An ensemble based approach using a combination of clustering and classification algorithms to enhance customer churn prediction in telecom industry"

_PeerJ Computer Science, doi:10.7717/peerj-cs.854_

## Round 0.1 · original submission · Major Revisions

Based on the comments from the reviewers, authors are required to make "major revision" and resubmit the paper.

The reviewers have suggested that you cite specific references. You may add them if you believe they are especially relevant. However, I do not expect you to include these citations, and if you do not include them, this will not influence my decision.

·

Basic reporting

The authors proposed churn prediction model is a hybrid model that is
based on the combination of clustering and classification algorithms using ensemble. First,
different clustering algorithms i.e K-means, K-medoids, X-means, and Random clustering
are evaluated individually on two churn prediction datasets. Then hybrid models are
introduced by combining clustering with seven different classification algorithms
individually and then evaluation is performed using ensembles. The author has done some good work but still, need improvements :

1- Abstract didn't contain much information about the proposed ensemble model. it contains not needed information as compared to meaningful information please update it.

2- there is no need to add a single sub-section 1.1 heading. just simply add a contribution.

3- Visualization is very poor.

4- In section 3.4.8 Ensemble classifiers author mentions ensemble classifiers voting, bagging, AdaBoost, and stacked without proper architecture discussion.

5- The flow of experiments should be added with diagrams.

6- Consider these ensemble architectures in literature and in comparison with your proposed model.


Rupapara, V., Rustam, F., Shahzad, H.F., Mehmood, A., Ashraf, I. and Choi, G.S., 2021. Impact of SMOTE on Imbalanced Text Features for Toxic Comments Classification using RVVC Model. IEEE Access.

Jamil, R., Ashraf, I., Rustam, F., Saad, E., Mehmood, A. and Choi, G.S., 2021. Detecting sarcasm in multi-domain datasets using convolutional neural networks and long short term memory network model. PeerJ Computer Science, 7, p.e645.

Rustam, F., Ashraf, I., Mehmood, A., Ullah, S. and Choi, G.S., 2019. Tweets classification on the base of sentiments for US airline companies. Entropy, 21(11), p.1078.

Rustam, F., Mehmood, A., Ullah, S., Ahmad, M., Khan, D.M., Choi, G.S. and On, B.W., 2020. Predicting pulsar stars using a random tree boosting voting classifier (RTB-VC). Astronomy and Computing, 32, p.100404.

Experimental design

The experimental design of the manuscript is not clear.
Please add a graphical representation of the proposed methodology.
Hyperparameter setting of models should be in tabular form.
How you tune your models, which method you used

Validity of the findings

Why author used Accuracy precision, recall, and f1 score for clustering method evaluation?
There is a big difference in accuracy and F1 score value which shows model overfitting please justify it.
Compare your proposed approach results with the above-mentioned studies.

Additional comments

Dear Editor
I hope you are doing well.
The manuscript is good needs some changes I have mention above
kind regards

·

Basic reporting

The authors proposed an approach for customer churn prediction. They proposed a hybrid model using the cluster and classification algorithms. The proposed research is evaluated on two different benchmark telecom data sets obtained from GitHub and Bigml platforms. The author needs improvement to make the manuscript significant.

How the author combines cluster and classification algorithms please describe it in more detail.
The combination of several models will increase the complexity so you work on the tradeoff between accuracy and efficiency.
Evaluate the clustering model in terms of RMSE, and MSE, MAE.
Why you selected these two datasets?
Apply some validation techniques to show the significance of the proposed approach in terms of standard deviation.
What is the impact of preprocessing on the model's performance?
Have you done some variation in preprocessing?
Apply state of the arts deep learning models LSTM, CNN GRU in comparison.
Update the literature review with recent studies.


Rupapara, V., Rustam, F., Shahzad, H.F., Mehmood, A., Ashraf, I. and Choi, G.S., 2021. Impact of SMOTE on Imbalanced Text Features for Toxic Comments Classification using RVVC Model. IEEE Access.
Rustam, F., Khalid, M., Aslam, W., Rupapara, V., Mehmood, A. and Choi, G.S., 2021. A performance comparison of supervised machine learning models for Covid-19 tweets sentiment analysis. Plos one, 16(2), p.e0245909.
Omar, B., Rustam, F., Mehmood, A. and Choi, G.S., 2021. Minimizing the overlapping degree to improve class-imbalanced learning under sparse feature selection: application to fraud detection. IEEE Access, 9, pp.28101-28110.
Rustam, F., Ashraf, I., Shafique, R., Mehmood, A., Ullah, S. and Sang Choi, G., 2021. Review prognosis system to predict employees job satisfaction using deep neural network. Computational Intelligence, 37(2), pp.924-950.

Experimental design

No comments

Validity of the findings

No comments

Additional comments

No comments

---

## Round 0.2 · accepted · Accept

Based on the reviewer comment for the revised version, the paper is accepted. Authors are advised to check for grammatical errors and typos.

·

Basic reporting

The revised version is good and the authors also resolve all my concerns, its good now for publication.

Experimental design

no comment

Validity of the findings

no comment

Additional comments

no comment

·

Basic reporting

Authors have done good work in revision and resolved all issues mentioned by me

Experimental design

no comment

Validity of the findings

no comment

Additional comments

no comment